# An inhalable nanoparticulate STING agonist synergizes with radiotherapy to confer long-term control of lung metastases

Yang Liu[1], William N. Crowe[1], Lulu Wang[1], Yong Lu[2], W. Jeffrey Petty[3], Amyn A. Habib [4] & Dawen Zhao[1,5]*

Mounting evidence suggests that the tumor microenvironment is profoundly immunosuppressive. Thus, mitigating tumor immunosuppression is crucial for inducing sustained antitumor immunity. Whereas previous studies involved intratumoral injection, we report here an inhalable nanoparticle-immunotherapy system targeting pulmonary antigen presenting cells (APCs) to enhance anticancer immunity against lung metastases. Inhalation of phosphatidylserine coated liposome loaded with STING agonist cyclic guanosine monophosphate–adenosine monophosphate (NP-cGAMP) in mouse models of lung metastases enables rapid distribution of NP-cGAMP to both lungs and subsequent uptake by APCs without causing immunopathology. NP-cGAMP designed for enhanced cytosolic release of cGAMP stimulates STING signaling and type I interferons production in APCs, resulting in the pro-inflammatory tumor microenvironment in multifocal lung metastases. Furthermore, fractionated radiation delivered to one tumor-bearing lung synergizes with inhaled NP-cGAMP, eliciting systemic anticancer immunity, controlling metastases in both lungs, and conferring long-term survival in mice with lung metastases and with repeated tumor challenge.

[1] Department of Biomedical Engineering, Wake Forest School of Medicine, Winston-Salem, NC 27157, USA. [2] Department of Microbiology and Immunology, Wake Forest School of Medicine, Winston-Salem, NC 27157, USA. [3] Department of Medicine, Section on Hematology and Oncology, Wake Forest School of Medicine, Winston-Salem, NC 27157, USA. [4] Department of Neurology and Neurotherapeutics, University of Texas Southwestern Medical Center and VA North Texas Medical Center, Dallas, TX 75390, USA. [5] Department of Cancer Biology, Wake Forest School of Medicine, Winston-Salem, NC 27157, USA. *email: dawzhao@wakehealth.edu

mmunotherapy is providing tremendous promise in the new era of cancer treatment. Checkpoint inhibitors and adoptive T cell transfer therapy have shown improved survival in melanoma, non-small-cell lung cancer, and renal cell cancer patients[1,2]. However, only a fraction of patients benefit from immunotherapy, and lack of specific tumor targeting is frequently associated with immune-related toxicity. Thus, many attempts are being made to improve anticancer immunity while reducing adverse effects. In contrast to systemic immunotherapy, intratumoral injection of immunomodulators is intended to focus the immune response locally on the malignancy and tumor draining lymph nodes (TDLNs)[3]. Moreover, given the heterogeneous nature of tumor antigens (TAs), intratumoral immunotherapy may have a potential for arousing a polyclonal antitumor immune response in situ against diverse cancer targets[3,4]. Among various types of immunomodulators, activators of the stimulator of interferon (IFN) genes (STING) pathway to elicit antitumor immunity have recently attracted much attention[5]. Being identified as a potent STING agonist, cyclic guanosine monophosphate–adenosine monophosphate (cGAMP) functions in the cytosol to ligate STING on endoplasmic reticulum (ER) membrane to activate STING pathway and type I IFN production[6,7]. Recent preclinical studies involving intratumoral injection of STING agonists alone or combined with irradiation (IR) have shown its aptitude for enhancing antitumor immunity[8–10]. Mechanistic studies indicate that activation of STING pathway within tumor-resident antigen-presenting cells (APCs) leading to type I IFNs production is indispensable for generation of adaptive immunity against tumors[11–13].

Although the in situ immunotherapy approach is attractive, there are several disadvantages associated with the intratumoral injection of immunostimulants. This approach is generally limited to those accessible tumors, and becomes even more challenging if repeated injections are needed. Structurally, cGAMP contains the phosphodiester bond that is susceptible to degradation by extracellular phosphodiesterase, also the two phosphodiester bonds in cGAMP restrict its penetration through the plasma membrane. Thus, in order to achieve adequate biological activity, cGAMP is commonly used at relatively high concentrations. However, excessive intratumoral cGAMP may induce programmed death-ligand 1 (PD-L1) overexpression on tumor cells and increase tumor-infiltrating regulatory T cells (Tregs), resulting in a negative impact on antitumor immunity[10,14–16]. Importantly, several studies have shown that intratumoral immunostimulants generally induce local immune response at the injected site, but have limited effect on distant, uninjected tumor sites, implying that the local approach may be inadequate to elicit systemic immunity, or that the systemic response even if induced, may be rendered inactive when exposed to the immunosuppressive tumor microenvironment (TME) at distant naive tumor sites[3,10,17].

To overcome these limitations, we set out to develop a nanotechnological strategy that enables targeted delivery of immunostimulants to intratumoral APCs. Here, we assemble phosphatidylserine (PS) on the surface of nanoparticle-cGAMP (NP-cGAMP) because membrane-exposed PS can be recognized and engulfed by macrophages and dendritic cells through their PS receptors[18,19]. Because of STING located on ER membrane, it is critical to ensure intracellular delivery and subsequent release of cGAMP to cytosol while bypassing lysosome. We thus utilize calcium phosphate (CaP) to precipitate cGAMP in the liposomal core. After ingestion by APCs, in response to low endosomal pH, cGAMP is released to the cytosol and binds to STING, initiating the STING pathway. We then demonstrate that inhalation of aerosolized NP-cGAMP enables rapid distribution of NP-cGAMP to individual lesions in both lungs bearing lung metastases and subsequent uptake by APCs to stimulate STING signaling and type 1 IFNs production (Fig. 1a). Last, we tested if inhaled NP-cGAMP can enhance radiotherapy in lung metastasis mouse models of B16-OVA melanoma and 4T1 breast carcinoma. Indeed, fractionated IR (8 Gy × 3) delivered to a lung bearing metastases combined with NP-cGAMP inhalation synergizes to control the metastases not only in the IR but also in the non-IR-treated lung. In addition to enhancing APC sensing of immunogenic IR at irradiated tumor sites and cross-presentation of TAs to prime effector T cells at TDLNs, inhaled NP-cGAMP promotes proinflammatory TME in non-irradiated tumors and facilitates recruitment of cytotoxic CD8$^+$ T cells, which contribute to robust anticancer immunity observed in this study (Fig. 1a). These data demonstrate that inhalation of NP-cGAMP may represent a pharmacological approach to enhance APC-mediated adaptive immune response against lung metastases.

## Results

**Preparation and characterization of PS-coated NP-cGAMP.** Liposomal NP-cGAMP was prepared in two steps using the water-in-oil reverse microemulsion method[20,21]. Both layers of liposome membrane are composed of anionic PS, of which PS exposed on the outer layer serves as the "eat me" signal for APCs, while the inner PS interacts with excessive cationic Ca$^{2+}$ of CaP to complex with cGAMP in the core. NP-cGAMP had an average diameter of 118.8 nm and a negative surface charge of −40.7 mV (Fig. 1b, c). High-performance liquid chromatography (HPLC) analysis indicated high encapsulation efficiency of cGAMP (71.9%) and high payload of cGAMP (31.6 μg/mg lipid). To study its stability, NP-cGAMP was incubated in 10% fetal bovine serum (FBS) at 37 °C. Dynamic light scattering showed that the size remained relatively unchanged for up to 120 h (Fig. 1d). Furthermore, to mimic physiological extracellular and acidic endosomal environments, drug release profiles of NP-cGAMP in phosphate-buffered saline (PBS) at pH 7.4, 6.5, 5.0 were determined (Fig. 1e). NP-cGAMP exhibited little cGAMP release at pH 7.4, but an acidic pH-responsive release (~40% at pH 6.5 and ~80% at pH 5.0 after 12 h). These data demonstrate that NP-cGAMP is stable at pH 7.4, but releases cGAMP in a pH-dependent manner.

**Preferential uptake of NP-cGAMP by APCs and STING activation.** To evaluate uptake of PS-coated NPs (PS-NPs) by APCs, DiR-labeled NP-DiR was incubated with various APCs including alveolar macrophage (AM), bone marrow-derived dendritic cells (BMDCs), and bone marrow-derived macrophage (BMDM). After 30 min incubation, robust uptake of NP-DiR by all three types of cells was observed (Fig. 2a). By contrast, there was minimal DiR signal in 4T1-luc breast cancer cells, B16F10 melanoma cells, or mouse vascular endothelial bEnd.3 cells (Supplementary Fig. 1a). To exclude the possibility that specific uptake by APCs was simply due to the anionic lipid coating, we reconstructed the liposome by replacing PS with anionic phosphatidic acid (PA) on the outer layer. Incubation with PA-NP (surface charge −42 mV) showed much less DiR in APCs compared to PS-NP (Supplementary Fig. 1b). We further investigated if the specific uptake by APCs is indeed PS-dependent. Our data clearly showed that the uptake of PS-NP by APCs was largely abolished if PS-NP was pretreated with anti-PS antibody (Ab) to block the surface PS (Supplementary Fig. 1c, d). We also demonstrated the uptake of NP-DiR by APC cells, but not cancer cells by flow cytometry (Supplementary Fig. 1e–h). These data indicate that PS-coated NPs are recognized and ingested by APCs in a PS-mediated process.

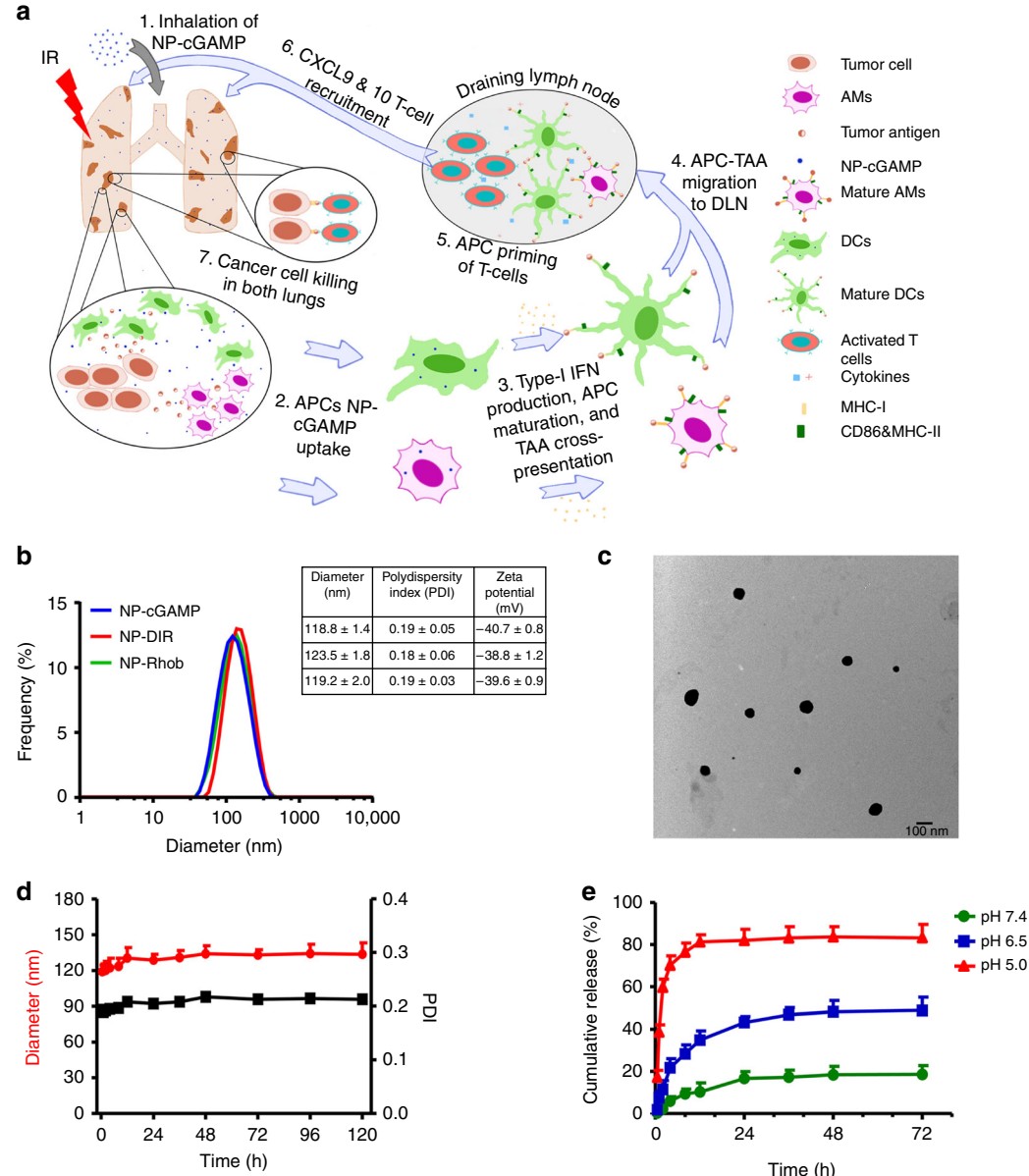

**Fig. 1** Characterization of PS-coated NP-cGAMP. **a** Schematic of the mode of action of the inhalable NP-cGAMP for enhancing antitumor immunity against lung metastases. Inhalation of PS-coated NP-cGAMP enables targeted delivery of the STING agonist, cGAMP to APCs in both irradiated and non-irradiated lung metastases. In addition to enhancing APC maturation, innate immune sensing of immunogenic IR, and cross-presentation of tumor antigen (TA) to prime effector T cells at the irradiated tumor sites, the inhaled NP-cGAMP promotes proinflammatory response in the non-irradiated tumors and facilitates recruitment of TA-specific effector T cells to lung metastases in both lungs, which contribute to therapeutic effects on both the IR and non-IR treated tumors. **b** Size distribution, PDI, and surface charge of NP-cGAMP, NP labeled with DIR (NP-DIR), and NP labeled with Rhob (NP-Rhob) by dynamic light scattering (DLS). **c** TEM images of NP-cGAMP. **d** Diameter change and PDI change of NP-cGAMP at pH 7.4 in 10% FBS (37 °C) over time. **e** Cumulative release of cGAMP from PS-coated NPs under pH 7.4, 6.5, or 5.0 at 0.5, 1, 2, 4, 8, 12, 24, 36, 48, and 72 h. Data are shown as mean ± SD of $n = 3$ biologically independent experiments. Source data are provided as a Source Data file

To assess whether NP-cGAMP can enhance cytosolic delivery of cGAMP to activate STING pathway and type I IFN production in APCs, BMDMs, BMDCs, and AMs were incubated with 100 nM free cGAMP or NP-cGAMP for 4 h. Relative expression of type I IFN and other inflammatory response genes were evaluated by real-time PCR. As shown in Fig. 2b, NP-cGAMP induced a drastic increase in expression of *Ifnb1* and *Ifna1*, as well as other proinflammatory genes, including *Tnf*, *Il1b*, *Il6*, *Il12b*, and *Cxcl9,10*, while only a modest increase was observed with free cGAMP (Fig. 2b). Consistent with the PCR results, enzyme-linked immunosorbent assay (ELISA) measurements detected a significant increase in the corresponding cytokines in the culture medium of APCs treated with NP-cGAMP (Supplementary Fig. 2). Western blot analysis revealed higher levels of phosphorylated TBK1 and IRF-3 in NP-cGAMP than free cGAMP-treated APCs (Fig. 2c), indicating activation of the STING pathway. These data demonstrate that NP-cGAMP enables efficient cytosolic delivery of cGAMP to activate the STING pathway and production of type I IFNs and other inflammatory cytokines in APCs.

**NP-cGAMP stimulates APC activation and cross-presentation.** The APCs that were treated with NP-cGAMP for 8 h were also

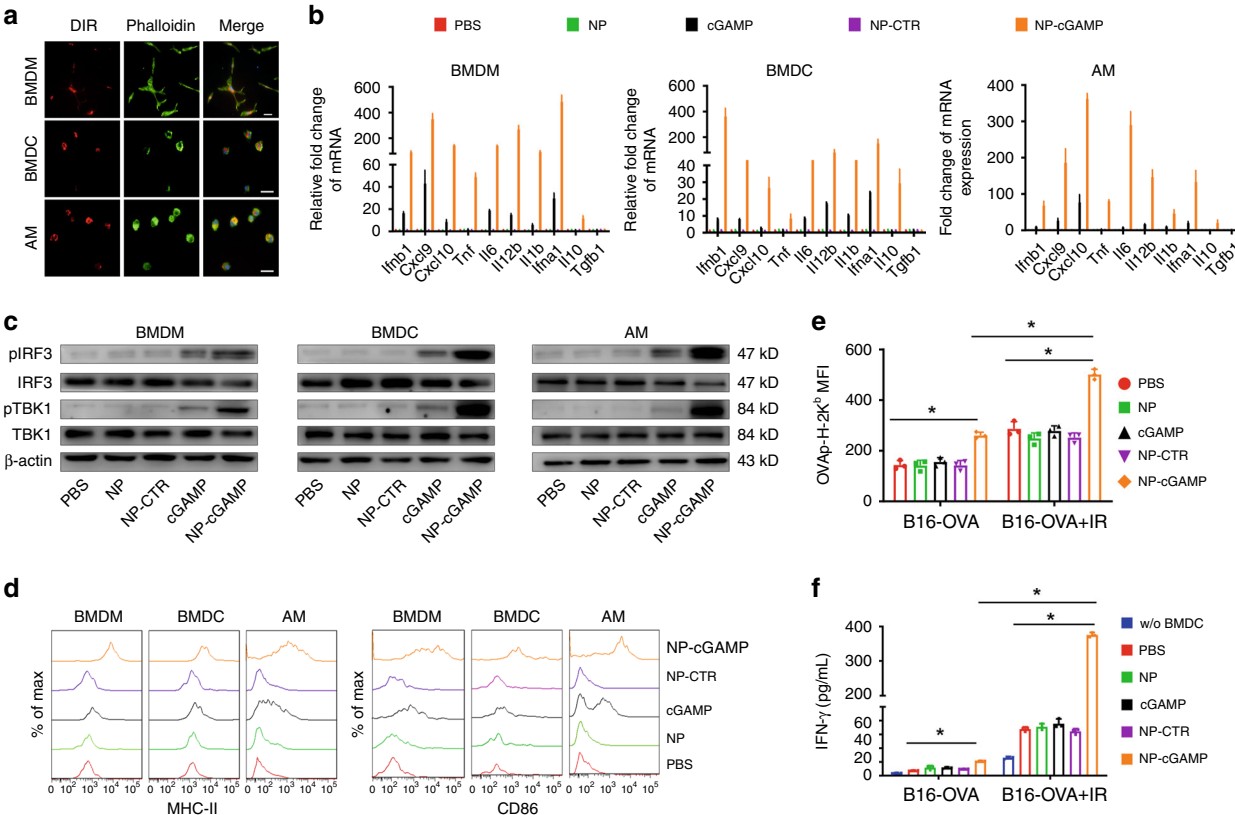

**Fig. 2** NP-cGAMP uptake by APCs activates STING pathway and CD8 T cell cross-priming. **a** BMDM, BMDC or AM cells were cultured with DiR-labeled PS-coated NPs (Red) for 30 min and fixed and co-stained with phalloidin (green) and DAPI (blue). Intracellular NP signals were clearly observed. Scale bar = 20 μm. **b** Real-time PCR of changes in messenger RNA (mRNA) levels of inflammatory cytokine genes in BMDM, BMDC, and AM after treated with free cGAMP (100 nM), NP-cGAMP (100 nM cGAMP), or NP-CTR (2′5′-GpAp as a control of cGAMP) for 4 h. **c** Western blot detection of STING pathway activation in BMDM, BMDC, and AM after treatment, as indicated, for 8 h. **d** FACS analysis of expression of the co-stimulatory molecule CD86 and MHC-II in BMDM, BMDC, and AMs after indicated treatment for 8 h. **e** B16-OVA cells treated with/without a single dose of 20 Gy IR were continued to culture for 72 h and then co-cultured with BMDCs under indicated treatment for 18 h. Expression of the OVA peptide SIINFEKL–MHC-I molecule Kb complex on the surface of BMDCs was analyzed by FACS. **f** ELISA assay of IFN-γ production from OT-1 CD8[+] T cells in vitro. CD8[+] T cells isolated from the OT-1 mouse spleen were added into the above mixture of BMDCs with IR- or non-IR-treated B16-OVA cells (BMDCs:T cells = 1:5) and further incubated for 18 h. IFN-γ concentrations in supernatant were determined by ELISA assay. Data were shown as mean ± SD of $n = 3$ biologically independent experiments. *$P < 0.001$ by Student's $T$ test. Source data are provided as a Source Data file

analyzed by flow cytometry (fluorescence-activated cell sorting (FACS)) to study the expression of major histocompatibility complex class II (MHC-II) and CD86. Marked increase (right shift) in both MHC-II and CD86 was observed in the NP-cGAMP-treated BMDM, BMDC, and AM (Fig. 2d). These data, together with previous observations of upregulated proinflammatory cytokines (Fig. 2b and Supplementary Fig. 2), suggest that NP-cGAMP stimulates APC maturation. To investigate if NP-cGAMP can enhance APC sensing and cross-presenting of TA to prime T cells, we then chose melanoma B16-OVA cells that stably express chicken ovalbumin (OVA) as a model antigen, and treated the cells with a single 20 Gy IR. Using anti-mouse SIIN-FEKL-2Kb Ab, we detected a significant increase in antigen presentation on the B16-OVA cells (Supplementary Fig. 3a, b). The B16-OVA cells with/without IR were then incubated with BMDCs in the presence of free cGAMP or NP-cGAMP for 18 h. FACS analysis showed that NP-cGAMP led to a significant increase in the OVA peptide–MHC-I complex on BMDCs ($p < 0.05$, Student's $T$ test), and even higher expression when BMDCs were incubated with the irradiated B16-OVA cells ($p < 0.001$; Fig. 2e and Supplementary Fig. 3c, d). Last, we isolated CD8[+] T cells from transgenic mice expressing T cell receptor specific for the OVA peptide SIINFEKL (OT-1) and placed the OT-1 CD8[+] cells into the above mixture of BMDCs and B16-OVA cells.

ELISA results showed significantly higher levels of IFN-γ in the culture medium where the BMDCs were preincubated with the irradiated B16-OVA in the presence of NP-cGAMP (Fig. 2f), indicating activation of TA-specific CD8[+] T cells. These data demonstrate that NP-cGAMP efficiently promotes APC activation and cross-presentation of TA to prime effector CD8[+] T cells.

**Targeted delivery of NP-cGAMP to lung APCs by inhalation.** To enable delivery of NP-cGAMP to deep lungs via inhalation, aerosolized NP-cGAMP was generated with a nebulizer system (Supplementary Fig. 4a). These aerosols had a mass mean aerodynamic diameter (MMAD) of 1.38 μm and geometric standard deviation (GSD) of 1.25 (Supplementary Fig. 4b), which falls into the size range optimal for deep lung deposition[22,23]. After inhaling NP-DiR, the mice with established 4T1-luc lung metastases were sacrificed at different times, and major organs were dissected and imaged ex vivo with IVIS. As shown in Fig. 3a, b, fluorescence signals were observed exclusively in lungs during the time course of 48 h. Concurring with the IVIS data, quantitative HPLC revealed predominant accumulation of the Rhod-b-labeled NPs in lungs with negligible amount in blood and other tissues (Fig. 3c). By applying various concentrations or durations of inhaled NP-Rhod-b, NP-Rhod-b concentrations in

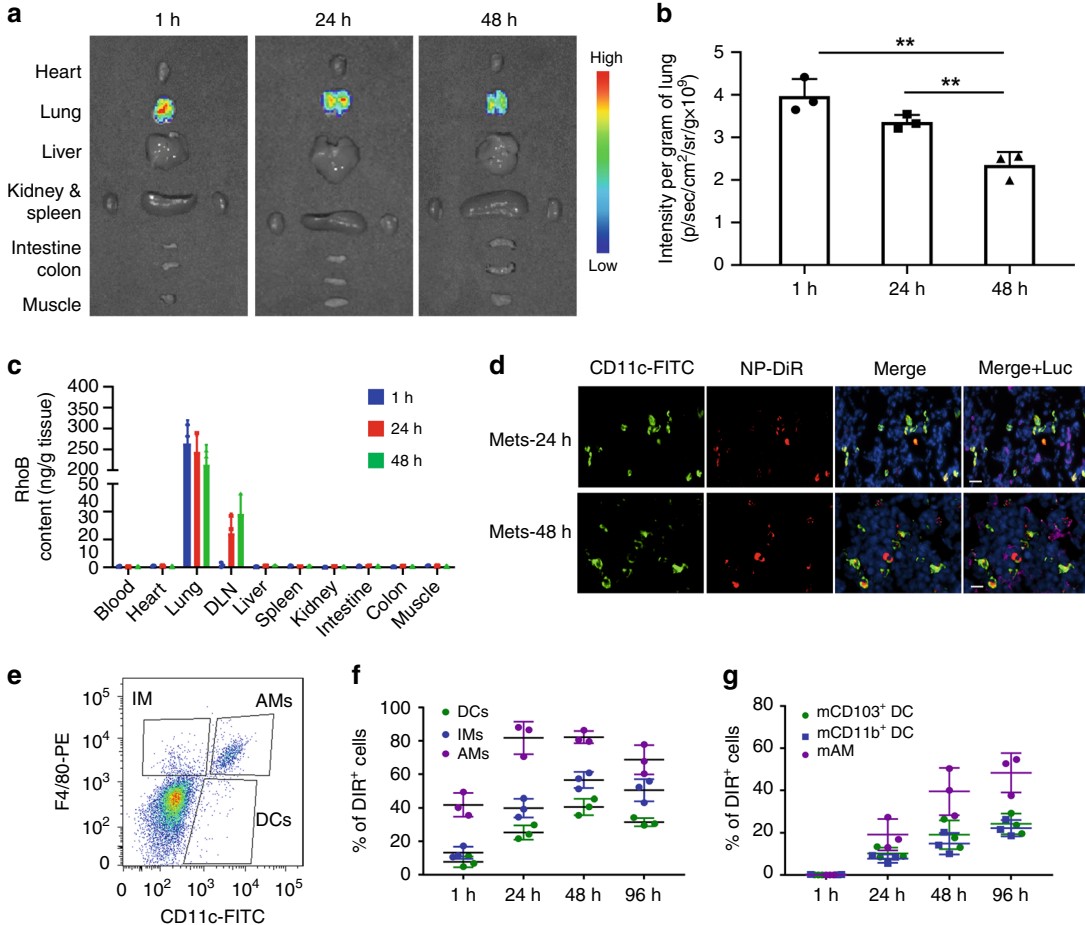

**Fig. 3** PS-NP via inhalation targets pulmonary antigen-presenting cells in lung metastases. **a** Representative ex vivo fluorescence imaging of major organs dissected from 4T1-luc lung metastases-bearing mice at 1, 24, and 48 h post inhalation of DiR-labeled PS-coated NPs. Light signals were exclusively from both lungs and **b** the lung signals were quantified ($n = 3$ biologically independent mice/time; **$p < 0.01$, Student's $t$ test). **c** HPLC measurements of concentrations of PS-coated NPs labeled with RhoB in various tissues of the 4T1-luc lung metastasis mice post inhalation ($n = 3$/time) were consistent with the IVIS imaging data. With the lung concentrations decreasing over time, NP increased in TDLNs. **d** Metastases-bearing lung tissues obtained 24 and 48 h post inhalation were subjected to immunofluorescent staining. The merged images clearly showed that DiR-labeled NPs (red) co-localized predominantly with CD11c+ APCs (green). Co-staining of 4T1 tumor cells with anti-luciferase (purple) indicated that the PS-NPs were distributed well into individual metastases and captured by intratumoral APCs. DAPI (blue), scale bar 20 μm. **e** Representative FACS characterization of pulmonary APC subsets in 4T1 lung metastasis-bearing lungs: alveolar macrophages (AMs; CD11c+F4/80+), interstitial macrophages (IMs; CD11c−F4/80+) and DCs (CD11c+F4/80−). **f** The percentage of DiR+ AMs, IMs, and DCs within their respective subpopulations was determined at 1, 24, 48, and 96 h after inhalation of NP-DiR. **g** The percentage of DiR+ migratory mAM, mDC103+, mCD11b + DCs in the total of APCs migrated to TDLNs was determined at different times post inhalation. Data are shown as mean ± SD of $n = 3$ biologically independent mice. Source data are provided as a Source Data file

metastases-bearing lungs were measured (Supplementary Fig. 4c), which was used to estimate the NP-cGAMP dose deposited to lungs. Under our experimental protocol with a 28 min inhalation of NP-cGAMP (37 μM cGAMP) in 5 mL PBS, we estimated 0.1 μg cGAMP deposited in an animal's lungs, less than one-hundredth of the dose of free cGAMP used for intratumoral injection in several studies[8,10]. Based on a previously published mathematical model[22] (Supplementary Fig. 4d), we can control the cGAMP dose delivered to lungs by changing inhalation duration or initial concentration of NP-cGAMP. Accuracy of the inhalation dose calculation was validated by direct measurements of cGAMP concentrations in tissues by HPLC using NP-cGAMP-FITC (Supplementary Fig. 4e).

It has been documented that the NPs become exposed to lung environment after they are delivered in aerosols to bronchioles and alveoli, at this point their physicochemical properties, for example, size and surface charge, most likely determine their fate[23]. We formulated NP-cGAMP with a mean diameter of ~120 nm because it is generally accepted within an optimal size range for a liposome drug delivery system after taking into account multiple factors such as payload and intratumoral diffusibility[24,25]. Anionic surface charge is also considered preferable for intratumoral distribution of liposome, while minimizing non-specific uptake by cell types other than APCs[26,27]. Indeed, immunohistochemical studies of lung tissues post inhalation of NP-DiR clearly showed that DiR signals were located primarily within CD11c+ APCs, and co-staining with anti-luciferase to label tumor cells further revealed that the NP-DiRs were distributed well intratumorally in individual metastases and engulfed by intratumoral CD11c+ APCs (Fig. 3d; Supplementary Fig. 5). Quantitative analysis showed that only a small fraction of NP-DiR was found outside APCs (<13%, Supplementary Fig. 5e). We further compared APC populations and DiR signals in metastases versus non-tumoral lung tissues. Our data revealed that there were significantly more APCs in tumors than non-tumoral lung tissue (Supplementary Fig. 5c), despite a slightly higher

percentage of APCs found to contain DiR-NP in the non-tumoral lung tissues (67% vs. 60% at 48 h; Supplementary Fig. 5f). Taken together, these data demonstrate the ability of PS-NPs to penetrate tumor tissues and target intratumoral APCs.

We also conducted FACS to quantify the uptake of NP-cGAMP by each subset of lung APCs. As shown in Fig. 3e, lung APCs were classified into AMs (CD11c$^+$F4/80$^+$), interstitial macrophages (IMs; CD11c$^-$F4/80$^+$), and DCs (CD11c$^+$F4/80$^-$). One hour post NP-DiR inhalation, $41.7 \pm 7.0\%$ (SD) of AMs, $13.2 \pm 3.6\%$ of IMs, and $7.6 \pm 3.2\%$ of DCs were DiR+, which increased over 48 h, and then decreased (Fig. 3f). Meanwhile, the lung-derived DiR+ mAMs and mCD103+ or mCD11b + DCs were found to increase over time up to 96 h in TDLNs (Supplementary Fig. 6; Fig. 3g), indicating the ability of these NP-captured lung APCs to migrate to TDLNs. CD103$^+$ DCs were sparsely populated in tumor-bearing lungs (Supplementary Fig. 7), but have been shown to play a potent role in cross-priming cytotoxic CD8$^+$ T cells in TDLNs[28,29]. Together, these data demonstrate that inhalation enables effective delivery of NP-cGAMP to intratumoral APCs in lung metastases.

**Inhaled NP-cGAMP synergizes with IR against lung metastases.** Radiotherapy is commonly used in clinic to treat lung cancer. In this study, we investigated if inhalation of NP-cGAMP can enhance radiotherapy against lung metastases. We established a melanoma lung metastasis model in immunocompetent mice by injecting B16-OVA cells intravenously. Five days later, after confirming multifocal lung metastases in both lungs, the mice were treated with IR alone, NP-cGAMP inhalation, or both. Fractionated IR (8 Gy × 3) was delivered to a specified area of the right lung while avoiding mediastinum and other normal tissues (Supplementary Fig. 8a–d). The 8 Gy × 3 dose schedule has previously been shown to induce immunogenic cell death and synergize with immunotherapy against breast cancer in mouse models[30,31]. Moreover, a recent study also reports that focal lung radiation with 8.5 Gy/fraction is safe without causing adverse effects[32]. For the combination treatment, NP-cGAMP was inhaled 24 h after each of three IR fractions. On day 18, the mice were sacrificed and treatment efficacy was analyzed by counting the number of metastases on the surface of lungs. As shown in Fig. 4a, for the mice treated with IR alone, there was a clear distinction between the irradiated right lung and the non-irradiated left lung, indicating that IR only affected the irradiated tumors. Despite decreased tumor volume, there was no significant change in the total number of lesions on the irradiated right lung, compared to the control treatment (Fig. 4b). NP-cGAMP inhalation alone led to a decrease in the number of metastatic foci in both lungs ($p < 0.05$, Student's $T$ test). NP-cGAMP plus IR achieved the highest therapeutic efficacy, inhibiting metastases in both the IR- and non-IR-treated lungs, and causing complete regression of lung metastases in some mice ($p < 0.001$, Fig. 4a, b). These data demonstrate that NP-cGAMP inhalation plus IR induces strong antitumor immunity that leads to regression of lung metastases in both the irradiated and non-irradiated lungs.

To study plausible mechanisms underlying the enhanced immunity, we first assessed if NP-cGAMP inhalation improved cross-presentation of TA in vivo. A subset of the mice from the above treatment groups was sacrificed 24 h after the last inhalation (48 h after the last IR). Both tumor-bearing lungs and TDLNs were dissected. Because CD103$^+$/CD8α$^+$ DCs have been implicated as the most competent APCs for cross-priming CD8$^+$ T cells in mice[33–35], we applied FACS gating strategies to differentiate CD103$^+$ DCs (CD103$^+$CD11b$^-$CD11c$^+$) from CD11b$^+$ DCs (CD11b$^+$CD103$^-$CD11c$^+$), and further analyzed the expression of the OVA peptide SIINFEKL–MHC-I complex

on these two types of DCs (Supplementary Fig. 7). As opposed to IR alone, which led to an increase only in the irradiated lung, IR plus inhalation induced significantly upregulated antigen presentation on CD103$^+$ DCs in both lungs (Fig. 4c), and the CD103$^+$ DCs with high antigen presentation were also detected in TDLNs (Fig. 4d), implicating migration of these APCs from tumor sites to TDLNs where they cross-prime T cells. Similarly, expression of SIINFEKL–MHC-I complex was detected on CD11b$^+$CD103$^-$ DCs (Supplementary Fig. 9g), which also increased after treatment with inhalation with/without IR. These data are consistent with previous reports that both types of DCs are capable of ingesting and processing TA and cross-presenting TA within the MHC-I complex[28,29]. However, CD103$^+$ DCs have been found to be more potent on cross-priming CD8$^+$ T cells, whereas CD11b$^+$ DCs may be involved in priming CD4$^+$ T cells through their MHC-II–peptide complex[28,29,35]. Consistent with our previous in vitro observations (Fig. 2d), NP-cGAMP inhalation activated the expression of co-stimulatory molecule, CD86, and MHC-II on APCs in both lungs (Supplementary Fig. 9). We next investigated whether NP-cGAMP inhalation with/without IR drove expansion of TA-specific T cells. As shown in Fig. 4e–g, significantly increased numbers of CD4$^+$ and CD8$^+$ T cells were observed in both lungs post IR, NP-cGAMP inhalation, or both. However, FACS analysis after SIINFEKL–MHC tetramer staining showed that the combination treatment led to a ~10-fold and >5-fold increase in the number of OVA-specific CD8$^+$ T cells in both lungs compared to the control and IR alone, respectively (Fig. 4h, i). Moreover, inhalation alone or combined with IR activated these tumor-specific CD8$^+$ T cells, evidenced by their higher levels of intracellular IFN-γ (Fig. 4j). Examinations of TDLNs from the combination treatment also revealed significant expansion of tumor-specific CD8$^+$ T cells (Fig. 5a, b). To interrogate if the combination treatment elicited systemic tumor-specific immunity, we examined spleens of the treated mice and found that there was indeed a significant increase in tetramer-positive CD8$^+$ T cells ($p < 0.001$, Student's $T$ test; Fig. 5a, c). We further conducted in vivo VITAL assay by injecting the carboxyfluorescein succinimidyl ester (CFSE) fluorescence-labeled OVA splenocytes into the previously treated mice (Supplementary Fig. 10). Compared with the relatively constant level of the non-OVA-labeled splenocytes, significantly more killing (>60%) of the OVA splenocytes was observed in the mice treated with IR plus inhalation ($p < 0.001$, Student's $T$ test; Fig. 5d, e), confirming induction of systemic tumor-specific immunity.

To determine whether NP-cGAMP induced anticancer immune response is APC-dependent and what subset of effector T cells is required to execute the response, we conducted depletion studies in the mice receiving IR plus NP-cGAMP. To deplete pulmonary APCs, we used the same nanoconstruct of PS-NP to encapsulate clodronate (NP-Clod) and delivered NP-Clod via inhalation 6 h before each of the three NP-cGAMP inhalations. To deplete CD4$^+$ or CD8$^+$ T cells, anti-mouse CD4 or CD8 antibodies were injected intraperitoneal (i.p.) 1 day before IR and repeated 7 days later. We found that depletion of lung APCs or CD8$^+$ T cells (Fig. 4b) significantly abrogated the antitumor function of the combination treatment, indicating that both APCs and CD8$^+$ T cells are required for the induced antitumor immunity. Depletion of CD4$^+$ T cells had no significant impact on the therapeutic response. These observations are in good agreement with previous reports, in which radiation alone or combined with STING agonists or other immunomodulators significantly enhance MHC-I expression and priming of cytotoxic CD8$^+$ T cell via activated APCs[8,10,36,37]. However, our current study cannot disentangle possible contribution of individual subsets of CD4$^+$ T cells, for example, the possibility of positive impact by eliminating Treg cells.

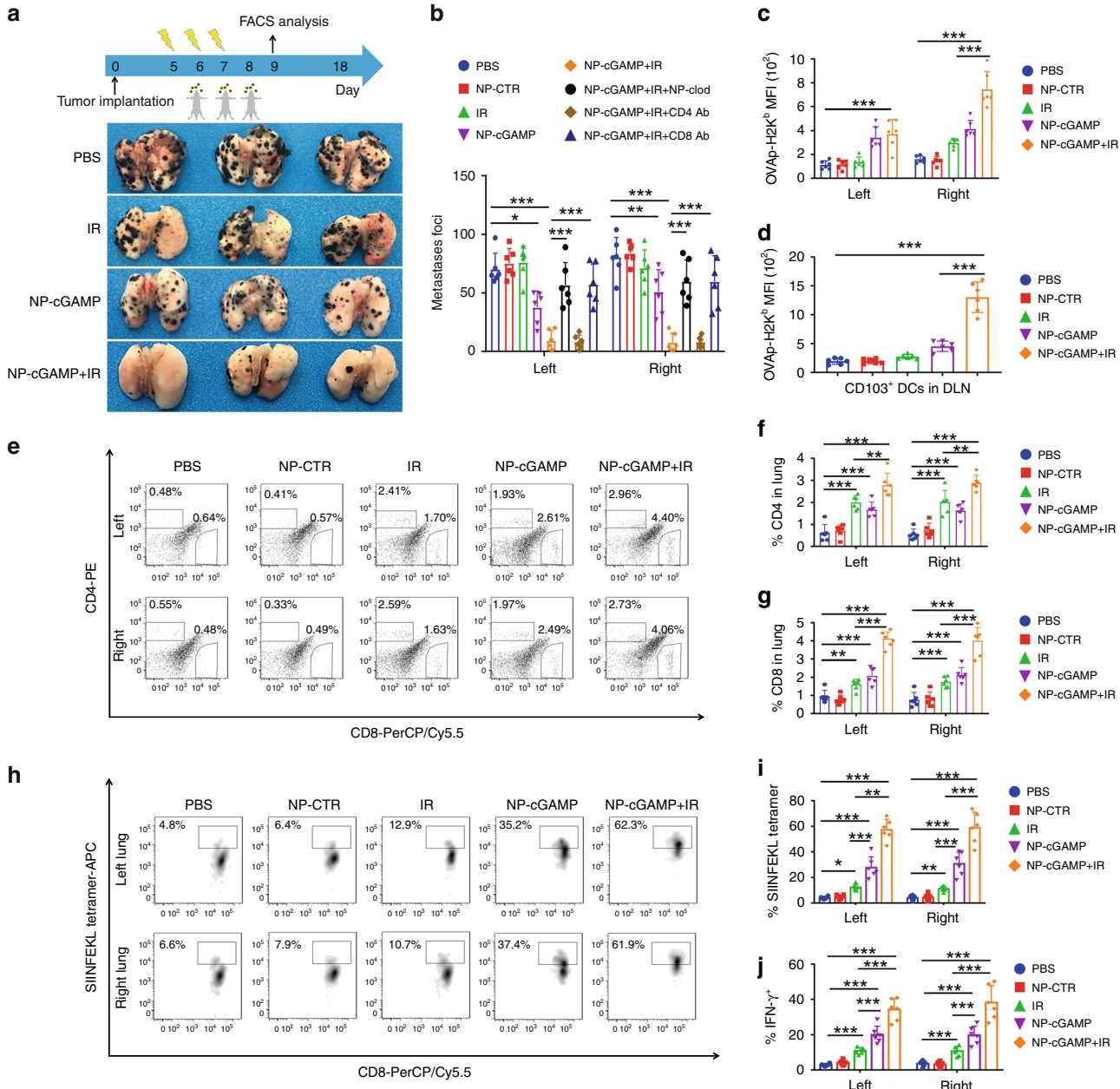

**Fig. 4** NP-cGAMP inhalation synergizes with radiotherapy by eliciting APC-mediated adaptive immunity. The B16-OVA melanoma lung metastasis model was established by intravenously (i.v. injecting $2 \times 10^5$ B16-OVA cells into C57BL/B6 mice. On day 5, after confirming development of multifocal metastases in both lungs, the mice were treated with fractionated radiation to the right lung (IR, 8 Gy × 3), inhalation of NP-cGAMP (24 h after each IR for three doses) or both. NP-CTR (2'5'-GpAp) served as a control of NP-cGAMP. To deplete pulmonary APCs, NP-clodronate was administered via inhalation 6 h before each of the three NP-cGAMP inhalations. To deplete CD4$^+$ or CD8$^+$ T cells, anti-CD4 Ab or anti-CD8α Ab was injected i.p. (400 μg), respectively, one day before IR and repeated 7 days later. **a** The mice ($n = 6$/group) were sacrificed on day 18 and representative lungs ($n = 3$) from the treatment groups were displayed. **b** Both lungs were examined under a dissecting microscope and a total of metastatic lung foci on each lung were counted. **c**, **d** Expressions of the complex of OVA peptide SIINFEKL–MHC-I molecule Kb on CD103$^+$ (CD11c$^+$CD103$^+$CD11b$^-$) DCs in the left, right lung and TDLNs ($n = 6$) were analyzed by FACS on day 9 (24 h after the last inhalation). **e–g** The number of CD8$^+$ and CD4$^+$ T cells in metastases-bearing left and right lung ($n = 6$) was quantified based on FACS. **h**, **i** FACS analysis indicated percentages of the SIINFEKL tetramer$^+$ CD8$^+$ T cells in the total of CD8$^+$ T cells in both lungs obtained on day 9. **j** Percentages of intracellular IFN-γ$^+$ SIINFEKL tetramer$^+$ CD8$^+$ T cells in the total of CD8$^+$ T cells were also quantified for the left and right lung in treatment groups as indicated. Data are shown as mean ± SD of $n = 6$ biologically independent samples. *$P < 0.05$, **$p < 0.01$, and ***$p < 0.001$ by Student's t test. Source data are provided as a Source Data file

Collectively, these data demonstrate that NP-cGAMP inhalation promotes APC immune sensing and cross-priming CD8$^+$ T cells, and synergizes with radiotherapy to elicit robust anticancer immunity that results in inhibition of both the irradiated and non-irradiated B16-OVA lung metastases.

**Inhalation of NP-cGAMP promotes proinflammatory TME.** It is well recognized that TME is extremely immunosuppressive, which may largely counteract the effect of antitumor immunity[3,38,39]. To investigate if NP-cGAMP inhalation improves the immunosuppressive TME, we analyzed the B16-OVA

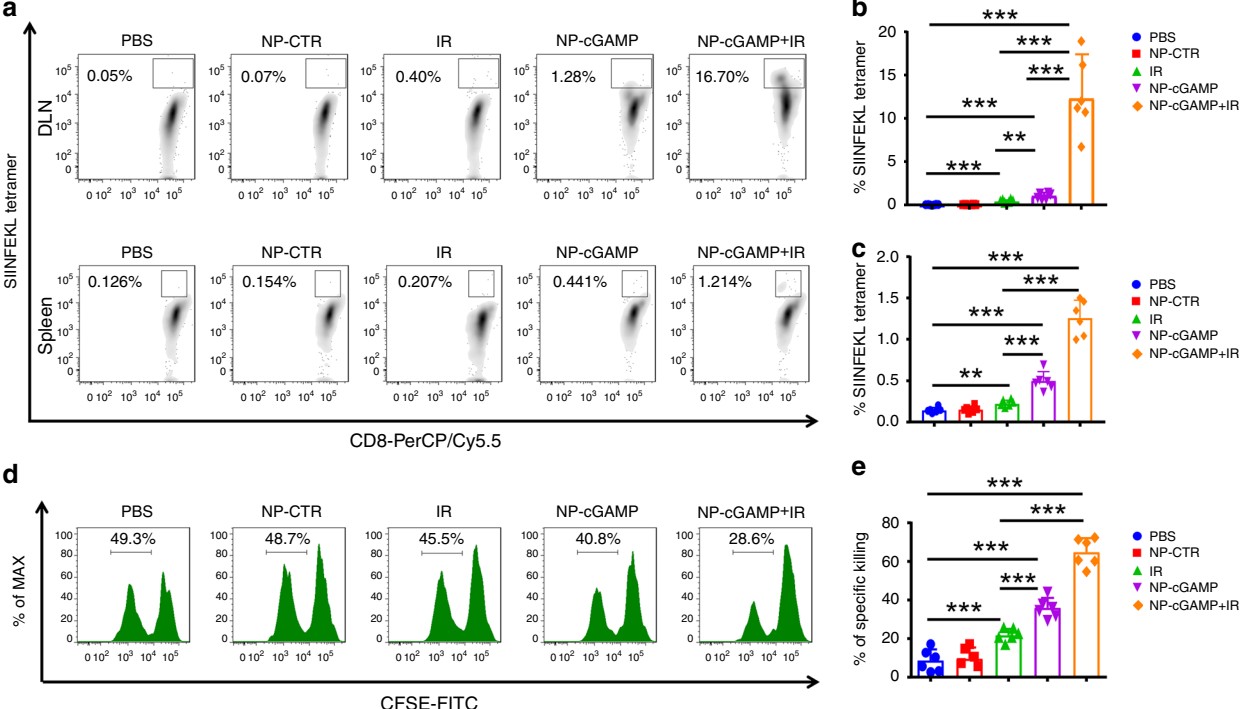

**Fig. 5** NP-cGAMP inhalation in combination with IR induces systemic tumor-specific immunity. **a** The lung TDLNs and spleen obtained on day 9 (24 h after the last inhalation) from the mice with indicated treatment were analyzed by FACS for SIINFEKL tetramer+ OVA-specific CD8 T cells. Frequency of SIINFEKL tetramer+ CD8+ T cells in TDLNs (**b**) and spleen (**c**). **d** In vivo VITAL assay. Spleen cells from naive C57BL/6 mice were isolated and half of the cells were pulsed with OVA$_{257-264}$ for 2 h in complete medium. The non-pulsed and OVA-pulsed cells labeled with high (0.5) or low (0.05) CFSE, respectively, were equally mixed and injected intravenously (i.v.) into the mice with indicated treatment (day 18 post tumor implant), and 16 h later blood was drawn for FACS analysis. Representative percentages of the low CFSE were indicated. **e** The combination treatment achieved maximal killings of the OVA splenocytes. Data are shown as mean ± SD of $n = 6$ biologically independent experiments. **P < 0.01 and ***p < 0.001 by Student's t test. Source data are provided as a Source Data file

metastases-bearing lung tissues from the previous treatment groups. ELISA results showed that NP-cGAMP inhalation led to significantly elevated levels of IFN-1β as well as tumor necrosis factor-α, IFNγ, interleukin-6 (IL-6), IL-12p40 in both lungs (Fig. 6a). By contrast, IR alone caused no significant change in the non-irradiated lung despite a moderate increase in the irradiated lung. Of interest, CXCL10, a ligand of CXCR3, significantly increased in both lungs after NP-cGAMP inhalation with/without IR (Fig. 6a), which could contribute to the marked increase in tumor-infiltrating T cells observed in both lungs[34] (Fig. 4e–g).

There is increasing evidence that the level of antitumor immunity is controlled by the balance of tumor-specific effector T cells and Tregs[14,40]. As presented previously in Figs. 4 and 5, IR alone was able to promote a moderate but significant increase in the number of tumor-infiltrating T cells in both lungs. However, IR also led to an increased number of tumor-infiltrating FoxP3+CD4+ Tregs, and consequently a significantly decreased ratio of CD8+ T/Tregs in the irradiated tumors ($p < 0.05$, Student's T test; Fig. 6b, c). Similar observations have been reported by others[10,41]. However, this negative effect was abrogated by NP-cGAMP inhalation, which significantly increased the ratio of CD8+ T/Treg in both lungs (Fig. 6b, c). These findings are in good agreement with several recent studies reporting activation of STING pathway or type I IFNs to directly inhibit co-stimulation-dependent Treg activation and proliferation[37,42]. Thus, unlike the direct intratumoral injection, our inhalation strategy enables delivery of NP-cGAMP to the lesions in both the irradiated and non-irradiated lungs to stimulate proinflammatory cytokines and suppress Tregs. The data further reiterate the necessity to overcome the

immunosuppressive TME not only in irradiated tumors but also non-irradiated tumors to elicit robust anticancer immunity.

**Efficacy is confirmed in 4T1 breast cancer lung metastases.** To determine whether potent antitumor immunity generated by IR plus NP-cGAMP inhalation was confined to B16-OVA lung metastases, we extended our study to 4T1 breast cancer lung metastases. To mimic clinical development of breast cancer lung metastases, we implanted 4T1-luc cells orthotopically into a mammary fat pad. When the tumor reached ~500 mm³ on day 14, the primary tumor was surgically removed. On day 18, after confirming establishment of lung metastases by bioluminescence imaging (BLI; Fig. 7a), the mice were randomly grouped and treated with IR alone (right lung), NP-cGAMP inhalation, or both, with the same dose and schedule as in the previous B16-OVA studies. Both BLI and magnetic resonance imaging (MRI) were applied to monitor growth of lung metastases post treatment. Compared to the control group, mice treated with IR or NP-cGAMP alone exhibited significantly lower BLI signals on days 32 and 39, indicating delayed tumor growth (Fig. 7a, b). IR plus NP-cGAMP inhalation led to further reduced BLI signals, even complete signal loss observed in some animals (Fig. 7a). MRI quantitatively evaluated tumor volume change by measuring individual lung metastases and summing to obtain the total tumor volume for each animal. The total tumor volume was significantly smaller in the combination group than IR or inhalation alone (Fig. 7c, d). The imaging data were consistent with ex vivo examination of lung metastases (Fig. 7e). For the long-term survival study, as presented in the Kaplan–Meier survival

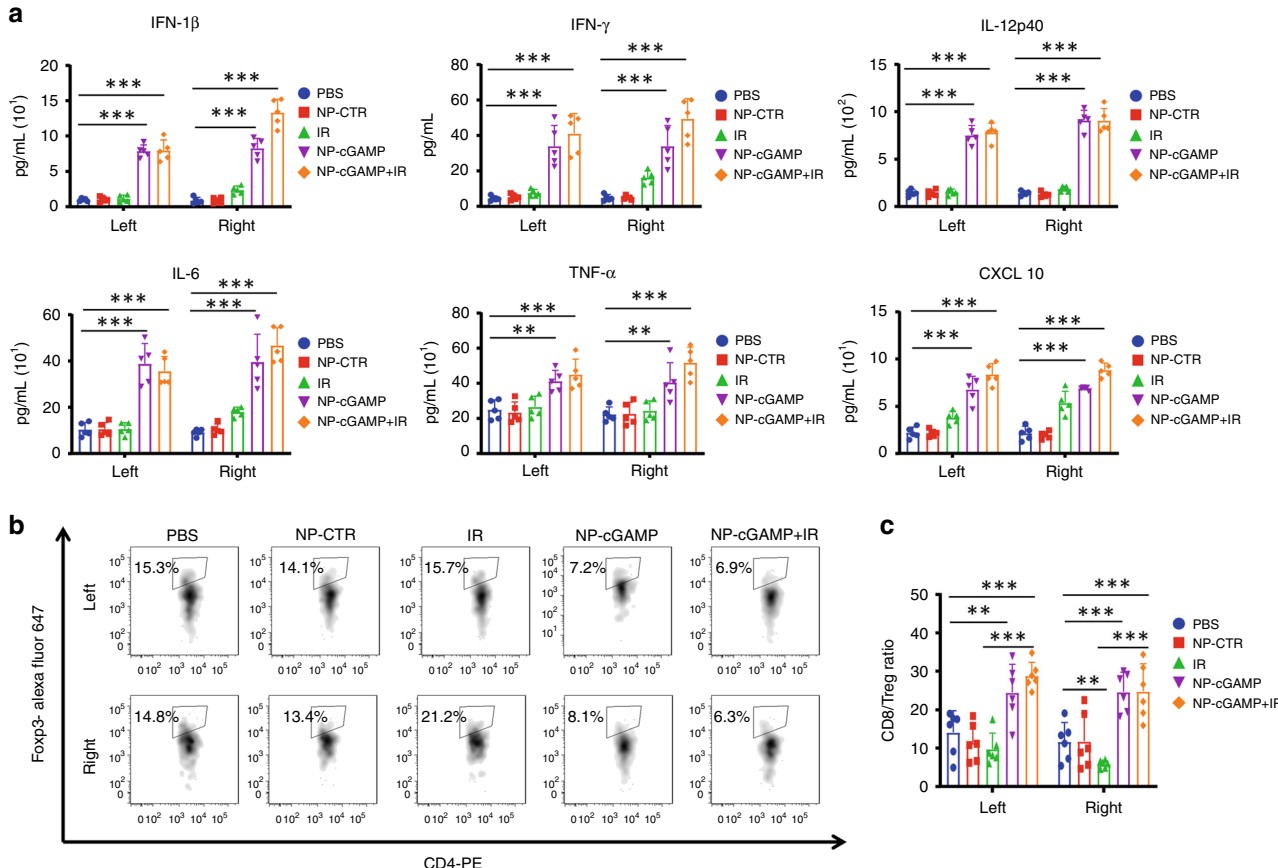

**Fig. 6** NP-cGAMP inhalation stimulates proinflammatory cytokines and improves the ratio of CD8⁺ T/Tregs. **a** The metastases-bearing lungs obtained on day 9 (24 h after the last inhalation) from the mice with indicated treatment were homogenized separately for the left and right (IR) lung. Various cytokines in the supernatant were measured by ELISA assay. Data shown as mean ± SD ($n = 5$). **b** FACS analysis of FoxP3⁺CD4⁺ regulatory T cells in the above metastases-bearing lung tissues and **c** quantitative data showed a significant increase in the ratio of CD8⁺ T cells/Tregs in both lungs in the inhalation alone or the combination treatment group, while a significant decrease in the ratio was detected in the irradiated right lung. Data are shown as mean ± SD of $n = 6$ biologically independent samples. **P < 0.01; ***p < 0.001 by Student's t test. Source data are provided as a Source Data file

curves, the mice treated with IR plus inhalation survived significantly longer ($p < 0.001$, log-rank test; Fig. 7f), 50% of them ($n = 4$) were completely cured, showing no sign of disease for at least 150 days (Fig. 7f). Similar to the B16-OVA study, we found that the survival benefit was abrogated after depletion of pulmonary APCs with NP-Clod (Fig. 7f), reiterating the indispensable role of APCs in the observed antitumor immunity. Moreover, the long-term surviving mice resisted secondary tumor challenge, indicating that the combination treatment triggers antitumor memory in this model (Fig. 7g).

Consistent with previous observations in the B16-OVA model, mechanistic studies revealed that inhalation of NP-cGAMP activated APCs in both 4T1-luc metastases-bearing lungs, as evidenced by significantly increased expression of CD86 and MHC-II on CD103⁺ DCs, CD11b⁺ DCs, and AMs (Supplementary Fig. 11). NP-cGAMP inhalation also led to drastic increase in IFN-1β and other proinflammatory cytokines in both lungs (Supplementary Fig. 12). Moreover, NP-cGAMP inhalation with/without IR increased the number of tumor-infiltrating T cells (Supplementary Fig. 13) and activated CD8⁺ T cells in both lungs (Fig. 8a, b). Like the B16-OVA, IR alone induced a significant increase in tumor-infiltrating FoxP3⁺CD4⁺ Tregs and a decrease in the ratio of CD8⁺ T/Tregs in the irradiated lung (Fig. 8c, d). However, NP-cGAMP inhalation offset the negative impact of IR by significantly increasing the CD8⁺ T/Treg ratio ($p < 0.01$, Student's T test; Fig. 8c, d). Immunohistochemical staining of

CD8⁺ T cells and FoxP3⁺ Tregs in lung metastases coincided with the FACS data (Fig. 8e, f). Together with previous studies in the B16-OVA model, NP-cGAMP inhalation plus IR effectively controls lung metastases.

To further investigate therapeutic efficacy of the combination treatment, we modified the above 4T1 lung metastasis protocol by retaining the primary tumor without surgical removal. In addition to the same combination lung treatment, the primary tumor was treated with/without intratumoral injection of NP-cGAMP. Our data showed that the combination lung treatment alone had modest effects on controlling the primary tumor (Supplementary Fig. 14a–d). However, in combination with intratumoral injection of NP-cGAMP (5 µg × 2), IR plus NP-cGAMP inhalation led to significant growth delay of the primary tumor and improved survival (Supplementary Fig. 14a–d). FACS analysis of the primary tumors clearly showed significantly increased APC maturation, tumor-infiltrating activated CD8⁺ T cells, and the CD8⁺/Treg ratio with the combination lung treatment plus intratumoral NP-cGAMP (Supplementary Fig. 14e–i). By contrast, lung treatment without intratumoral NP-cGAMP induced essentially no change in the TME of primary tumor even though increased activation of effector CD8⁺ T cells was evident in spleens of these mice (Supplementary Fig. 14e–j). These results are in line with a number of recent publications[43,44]. Chao et al.[43] reported that treatment of radiation plus intratumoral CpG on primary 4T1 tumor had modest effects on lung metastases, whereas an addition of systemic

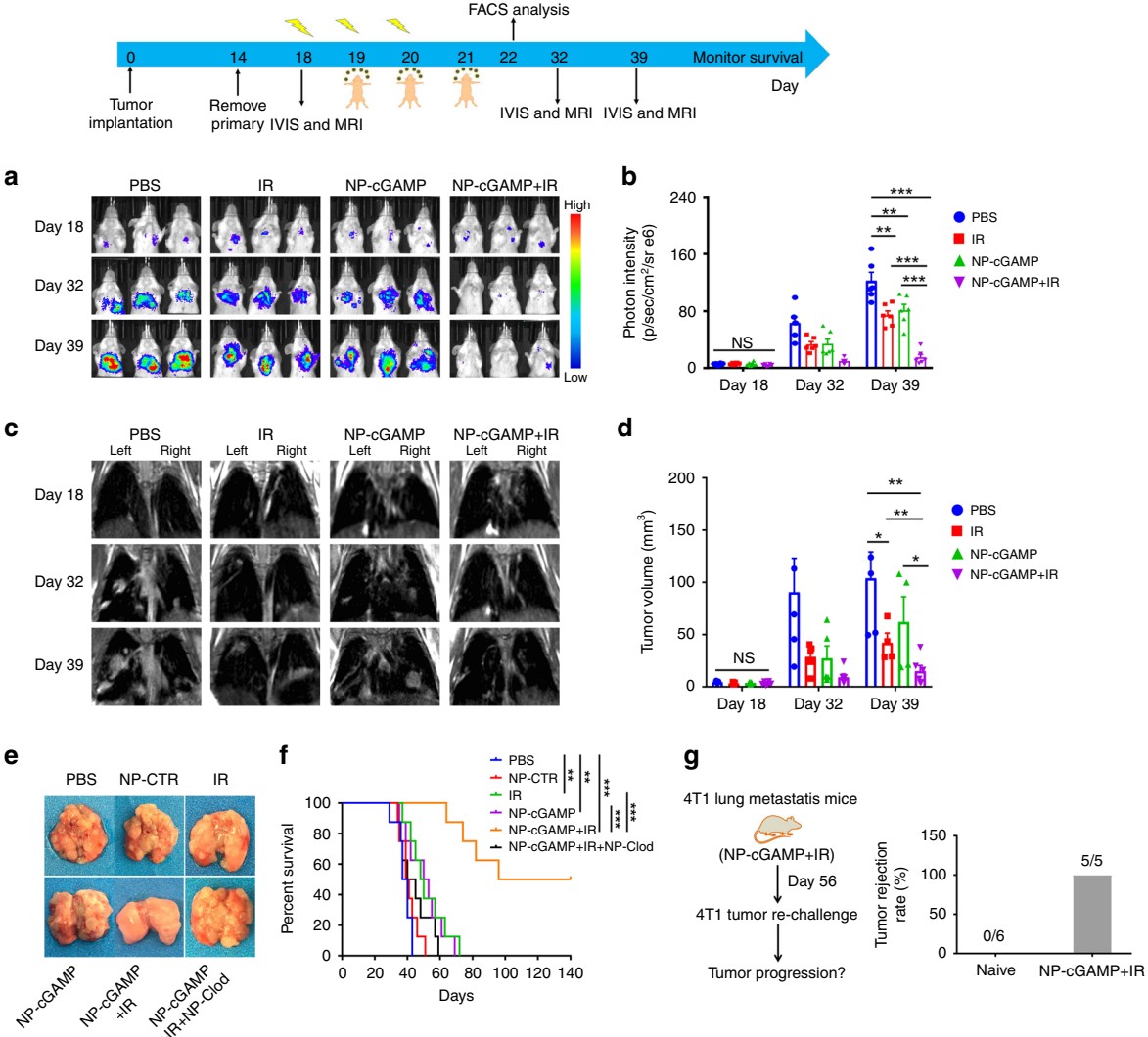

**Fig. 7** NP-cGAMP inhalation plus IR significantly is efficacious against 4T1 breast cancer lung metastases. Female BALB/c mice were implanted orthotopically with $1 \times 10^6$ 4T1-Luc cells into the fourth mammary fat pad. On day 14 when the tumors size reached around 500 mm³, the primary tumor was removed. On day 18, after confirming establishment of lung metastases by visualizing bioluminescence imaging (BLI) signals in the chest, the mice were treated with fractionated radiation to right lung (IR, 8 Gy ×3), inhalation of NP-cGAMP, or both. NP-cGAMP was inhaled 24 h after each IR for a total of three inhalations. Longitudinal BLI and MRI were conducted to monitor growth of lung metastases on days 18, 32, and 39. **a** IVIS images of three representative animals from each treatment group and **b** quantitative BLI light intensity of the chest. Data were shown as mean ± s.e.m. of n = 6 biologically independent mice. **c** Longitudinal MRI T2-weighted imaging follow-up of a representative mouse chest from each treatment group and **d** MRI quantitative tumor volume. Data were shown as mean ± s.e.m. of 4–5 mice. *P < 0.05; **p < 0.01; ***p < 0.001 by Student's T test. **e** Ex vivo lung image represented from each treatment group on day 39. Inhalation of NP-CTR (2′5′-GpAp as a control of cGAMP) for three doses; an additional group with NP-cGAMP plus IR, in which NP-clodronate was administered via inhalation 6 h before each of the three NP-cGAMP inhalations to deplete pulmonary APCs. **f** Kaplan–Meier survival curves of the treatment groups up to 140 days after tumor (n = 8/group) were plotted and statistically analyzed by log-rank test, *p < 0.05; **p < 0.01; ***p < 0.001. **g** Mice cured with NP-cGAMP + IR were re-challenged 56 days later with 4T1-Luc cells. Naive mice challenged at the same time served as positive controls. Data showed the percent of mice rejecting 4T1-Luc tumor re-challenge in each group. Source data are provided as a Source Data file

anti-CTLA4 Ab regressed lung metastases and prolonged survival[43]. Immune checkpoint inhibitors are known for their ability to activate tumor-infiltrating effector T cells, which are otherwise exhausted in the TME of the distant non-treated tumor. Taken together, these data support our hypothesis that the immunosuppressive TME negatively impacts anticancer immunity, and it is indispensable to overcome it to elicit durable antitumor immunity.

**Inhalation of NP-cGAMP is safe.** Liposome is biocompatible and safe as most clinically approved NP drugs are formulated in liposome. Inhalation of NP-cGAMP was well tolerated by unanesthetized mice. To investigate whether NP-cGAMP inhalation potentially induces systemic immune toxicity, we conducted systematic studies by monitoring body weight, measuring cytokine levels and liver enzymes in blood and examining major organs histopathologically in both healthy mice and lung metastases-bearing mice. In the healthy cohort, there was no significant difference in body weight, liver enzymes aspartate aminotransferase and alanine aminotransferase or cytokines in blood between control treatment and inhalation with/without IR at day 1 or 22 post treatment (Supplementary Fig. 15). Similarly, no significant change was detected in lung metastases-bearing mice (Supplementary Fig. 16). Hematoxylin and eosin (H&E)

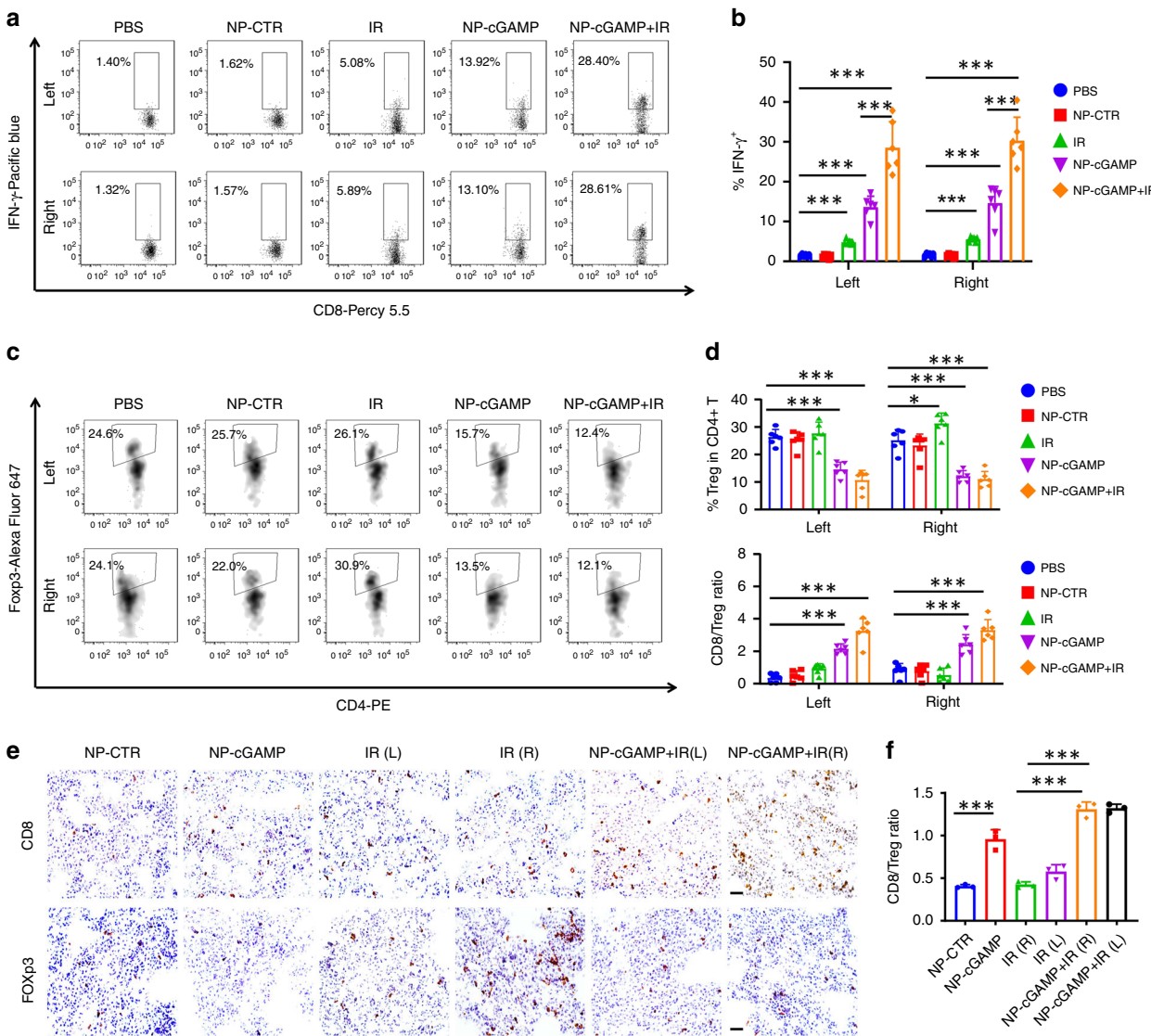

**Fig. 8** NP-cGAMP inhalation activates CD8 T cells and improves the ratio of CD8$^+$ T/Treg in 4T1 lung metastases. Twenty four hours after the last inhalation, the mice under indicated treatment ($n = 6$ biologically independent mice/group) were sacrificed and both metastases-bearing lungs were dissected for FACS and immunohistochemical studies. **a** After ex vivo re-stimulation of T cells with phorbol 12-myristate 13-acetate/ionomycin, representative FACS dot plots ($n = 5000$) of activated CD8$^+$ T cells by staining intracellular IFNγ from the left and right (IR) lung of each indicated treatment. **b** Quantitative analysis of frequency of IFN-γ$^+$CD8$^+$ T cells in the left and right lung. **c** Representative FACS analysis and **d** quantitative frequency of FoxP3$^+$CD4$^+$ Treg cells and ratio of CD8/Treg. **e** Immunohistochemical staining of CD8$^+$ T cells and FoxP3$^+$ Tregs in metastases-bearing lung tissues of $n = 3$ biologically independent mice under indicated treatment, and quantitative data showing the mean ± SD ratio of CD8/Treg in lung metastases. **f** NP-CTR: NP-2′5′-GpAp as a negative control of NP-cGAMP; IR (L) and IR (R): left (non-irradiated) and right (irradiated) lung of IR alone; the same applies to NP-cGAMP + IR(L) or IR(R). Scale bar = 20 μm. *$P < 0.05$; ***$p < 0.001$ by Student's $T$ test. Source data are provided as a Source Data file

staining of major organs at day 1 or 22 showed no visible morphological change in healthy or lung metastases-bearing mice treated with inhalation with/without IR (Supplementary Fig. 17). To investigate whether the treatment caused any short- or long-term lung toxicity, we first examined the healthy mice on days 1 and 22 post treatment. H&E staining showed no significant morphological change, for example, hemorrhage, despite some increased infiltration of cells, likely leukocytes, observed in the lung treated with IR plus inhalation on day 1. On day 22, there was no sign of inflammation in the lung (Supplementary Fig. 18a, b). We then studied the four mice that were cured with the combination treatment. After sacrifice on day 150, visual examinations of major organs, including lungs, liver, and kidney, revealed no obvious lesions; histology found no microscopic

lesions in lungs, confirming no residual metastases (Supplementary Fig. 18c). The shape of alveoli, alveolar wall thickness, and microvasculature in the non-irradiated left lung all looked similar to their healthy counterparts. While a few small regions in the irradiated lung were seen to contain clustering macrophage-like cells and some slightly thickened alveolar walls (Supplementary Fig. 18c), there was no obvious pathological change, for example, fibrosis. The lack of local and systemic toxicity may be attributed to the APC-targeting ability of NP-cGAMP and the low dose used for inhalation, which led to negligible amounts of cGAMP taken up by other lung cells (Fig. 3d; Supplementary Fig. 5) or entering blood (Fig. 3a–c). Nevertheless, NP-cGAMP inhalation in combination with IR is safe and effective against lung metastases.

## Discussion

Radiation is known for its potential to kill cancer cells to release TA[36,43,45–48], which may trigger systemic immunity against cancer metastases[31,48,49]. However, radiation effect on non-irradiated tumors is rarely seen in clinic. In this study, we demonstrate an inhalable NP-immunotherapy system for APC-targeted delivery of STING agonists to activate intratumoral APCs and enhance immune sensing of immunogenic radio-therapy in lung metastases mouse models. NP-cGAMP inhalation synergizes with fractionated IR to generate potent antitumor immunity against both the irradiated and non-irradiated lung metastases. The ability of NP-cGAMP inhalation to remodel the TME by converting "cold" into "hot" tumors in both the irra-diated and non-irradiated tumors contributes to the therapeutic efficacy.

Intratumoral injection of immunomodulators to elicit anticancer immunity has shown promising results in preclinical studies and is being investigated extensively in clinical trials. Although we also demonstrated the utility of intratumoral injection of NP-cGAMP, we were not intending to compare it with the inhalation approach in this study, because the individuals who would receive this therapy come from different patient populations. In the context of clinical cancer development and treatment strategy, primary cancer, for example, breast cancer or melanoma, is commonly resected by surgery or locally treated once detected. However, lung metastases may develop years later in these patients, and in many cases when the primary cancer is well controlled. We are specifically considering the potential of our inhalation approach for treating this population of cancer patients. From a clinical perspective, these patients have the highest mortality, thus development of effective treatment, for example, immunotherapy, is the most needed.

Given that lung is an extremely common site for breast cancer and melanoma to metastasize[50,51] and in many cases multiple lesions develop at peripheral lung[52,53], the inhalation approach, which has advantages including its non-invasiveness, feasibility for repeated procedures, and accessibility to multiple lung lesions/lobes at the same time, may have particular translational rele-vance to deliver immunomodulators to these lung metastases. With a similar strategy of using hypofractionated stereotactic body radiation therapy to treat a single lesion or few lesions in a lung[54], and combined with NP-cGAMP inhalation to activate antitumor immunity in both irradiated and non-irradiated tumors, we believe that this nano-immunotherapy system may have the potential for treating lung metastases arising from a variety of primary cancer types.

## Methods

**Preparation of PS-coated NP-cGAMP**. The liposomal NP-cGAMP was prepared in two steps using the water-in-oil reverse microemulsion method[20,21]. DSPC (1,2-distearoyl-sn-glycero-3-phosphocholine,18:0 PC), cholesterol, brain PS (L-α-phos-phatidylserine), DSPA (1,2-distearoyl-sn-glycero-3-phosphate,18:0 PA), and Rhod-b (18:1 Liss Rhod PE, 1,2-dioleoyl-sn-glycero-3-phosphoethanolamine-N-(lissa-mine rhodamine B sulfonyl)) were purchased from Avanti Polar Lipids. cGAMP (2′3′-cGAMP, cyclic [G(2′,5′)pA(3′,5′)p]) and cGAMP control (2′5′-GpAp) were purchased from InvivoGen. Both layers of the liposome membrane are composed of anionic PS. Briefly, 150 μL CaCl₂ (2.5 M in ddH₂O, pH = 7.0) was added to 5 mL mixed oil phase 1 (cyclohexane:igepal co-520 = 80:20) in 50 mL flask and stirred at 600 r.p.m. for 20 min to form a well-dispersed microemulsion phase. Oil phase 2 was prepared by adding 50 μL lipid mixture (20 mM, PS:DSPC:cholesterol = 5:4:1) to 5 ml mixed oil phase (cyclohexane:igepal co-520 = 80:20). A similar micro-emulsion containing sodium phosphate was prepared by adding 150 μL Na₂HPO₄ (25 mM in ddH₂O, pH = 9.0) to 5 ml mixed oil phase 2 with the calcium (Ca): phosphate (P) ratio of 100:1. The CaP core with single layer anionic lipid coating was formed by mixing oil phase 1 and oil phase 2 and stirring at 600 r.p.m. for 20 min. To collect the lipid-coated CaP cores, 10 ml ethanol was added to the mixture and centrifuged at 16,000 × g for 15 min, followed by washing with ethanol three times. The collected CaP cores were dispersed in 1 ml chloroform and cen-trifuged at 2000 × g for 5 min to remove CaP precipitates without lipid coatings. The supernatants containing a single layer of lipid-coated CaP were further mixed

with 70 μL lipid mixture (20 mM, brain PS:DSPC:cholesterol = 5:4:1, molar ratio, in CHCl₃) into 50 ml flask, followed by chloroform evaporation under reduced pressure to form a lipid film. NP-CaP NPs with bilayer lipid coating were formed by adding 1 ml PBS (0.1 M, pH = 7.4) and rehydrating under water bath sonication for 5 min and sonic probe at 20 W for another 2 min at 70 °C. The resulting NPs were further filtered with 0.45 μM membrane to remove the free lipid aggregates and stored at 4 °C. For DSPA-exposed CaP NPs, DSPA at 20 mM was used instead of brain PS during the preparation. To load cGAMP, half of the desired content of cGAMP was mixed with CaCl₂ solution and Na₂HPO₄. For fluorescently labeling NPs, 18:1 Liss Rhod PE (1,2-dioleoyl-sn-glycero-3-phosphoethanolamine-N-(lis-samine rhodamine B sulfonyl); Rhod-b) was added to the second lipid mixture with a molar ratio of 1%. DiR was further used for labeling the NPs by adding DiR directly to the second lipid mixture at a molar ratio of 5%.

**Characterization of NP-cGAMP**. The size, size distribution, and zeta potential of NP-cGAMP in aqueous solution were measured by a Malvern Zetasizer Nano ZS90. Transmission electron microscopy (TEM) measurements were performed on an FEI Tecnai Bio Twin TEM. To determine cGAMP loading efficiency, 0.2 ml of NPs were incubated with 0.2 ml 0.8 M HCl for 24 h to dissolve CaP NPs and release cGAMP. The mixture was further centrifuged at 14,000 × g for 15 min, and the supernatant was gathered to quantify the cGAMP concentrations by HPLC ana-lyses using an Agilent 1100 HPLC system. The release of cGAMP from NPs was assessed by the dialysis of NP-cGAMP solution against release medium at different pH (pH 7.4, 6.5 and 5.0). The release medium was removed for analysis at 0.5, 1, 2, 4, 8, 12, 24, 36, 48, and 72 h. cGAMP content in the release medium was deter-mined by HPLC. The NPs were incubated in PBS (pH 7.4, 0.01 M) with 10% FBS (v/v) at 37 °C for 5 days to study particle stability. The change in particle size was monitored at specific time intervals by Zetasizer.

**Cell lines**. The OVA-transfected mouse melanoma B16F10 cells (B16-OVA) and parental B16F10 cells (ATCC), and the 4T1 breast cancer cells stably transfected with firefly luciferase (4T1-luc; provided by Dr. David Soto Pantoja, Wake Forest), mouse vascular endothelial cells, bEnd.3 (ATCC) were cultured in Dulbecco's modified Eagle's medium (DMEM), supplemented with 10% FBS, 100 U/mL penicillin, and 100 μg/mL streptomycin and maintained in a humidified atmo-sphere containing 5% CO₂ at 37 °C.

**Induction of BMDM and BMDC and isolation of AM cells**. BMDMs were generated by culturing bone marrow cells flushed from C57BL/6 mouse femurs. Briefly, the gathered bone marrow cells were incubated in completed α-MEM (minimum essential medium Eagle-alpha modification) containing 10% heat-inactivated FBS and penicillin/streptomycin. After 18 h, the floating cells were collected and cultured in complete BMDM medium (complete α-MEM supple-mented with 50 ng/mL M-CSF). Three days later, the adherent cells were used as macrophages and phenotyped by determining the expression of CD11b and F4/80 (typically 70–85% CD11b⁺F4/80⁺).

BMDCs were also generated by culturing bone marrow cells flushed from the femurs of C57BL/6 mice in BMDC medium: RPMI-1640 containing 10% heat-inactivated FBS, penicillin/streptomycin, 20 ng/mL granulocyte–macrophage colony-stimulating factor, 5 ng/mL IL-4, and 1× 2-mercaptoethanol. The culture medium was half-replaced every 2 days, and the non-adherent and loosely adherent immature DCs were collected on day 8 and phenotyped by determining the expression of CD11c (typically 60–80% CD11b⁺).

To obtain AMs, C57BL/6 mice were sacrificed by CO₂. The trachea was then cannulated with a blunt 22-gauge needle and tied in place. Bronchoalveolar lavage was flushed with PBS containing 0.5 mM EDTA. Aliquots of 1 mL were instilled into the lungs and aspirated back into the syringe. This procedure was repeated three times per mouse. The gathered cells were cultured in DMEM medium with 10% heat-inactivated FBS, 2-mercaptoethanol (1×), and penicillin/streptomycin.

**Cellular uptake of PS-coated NPs in vitro**. Briefly, 5000 BMDM, BMDC, and AM cells were seeded in 24-well plates with poly-lysine-coated coverslips for 24 h. Then, the media were replaced with the complete medium containing PS-exposed or DSPA-exposed NPs labeled with DiR and cultured for another 0.5 h. For the blocking study, PS-NPs were pretreated with/without anti-PS antibody PGN635 (25 μg/mL; Peregrine Pharmaceuticals, Tustin, CA) for 2 h before incubating with AM or BMDC cells for 30 mins[55]. Cells were then washed twice with cold PBS and fixed with 4% formaldehyde for 15 min. After 1% Triton X-100 treatment for 15 min, the cells were stained with Alexa Fluor 488 phalloidin (Thermo Fisher) for 30 min and incubated with 1 μg/mL DAPI (4′,6-diamidino-2-phenylindole) for 5 min. The cellular uptake of NPs was observed by fluorescence microscope. To determine whether the PS-coated NPs are preferentially recognized and ingested by APCs, we repeated the above experiment with 4T1-luc or B16F10 cancer cells or normal mouse vascular endothelial bEnd.3 cells.

**Real-time quantitative PCR of type I IFN and other inflammatory genes**. BMDM, BMDC, or AM cells (10⁵) were seeded into 6-well plates and cultured for 24 h. The cells were then incubated at 37 °C with free cGAMP or NP-cGAMP (100 nM cGAMP) suspended in complete culture medium for 4 h. All cellular RNA

was collected using TRIzol reagent (Thermo Fisher). One microgram of total RNA was transcribed into complementary DNA (cDNA) using High-Capacity cDNA Reverse Transcription Kit (Thermo Fisher), and real-time quantitative PCR was performed using PowerUp SYBR Green Master Mix (Thermo Fisher) with primers listed in Table S1. Messenger RNA levels were normalized against the house-keeping gene *GAPDH* (glyceraldehyde 3-phosphate dehydrogenase).

**Western blot of STING pathway activation by NP-cGAMP.** For western blot analysis, the above APC cells incubated with free cGAMP or NP-cGAMP (100 nM cGAMP) for 8 h were washed and lysed in 50 μL of lysis buffer containing a protease inhibitor cocktail (Roche). The cell lysates with total protein (40 μg) were electrophoresed. Anti-phospho-IRF-3 (1:1000), anti-IRF-3 (1:1000), anti-phospho-TBK1 (1:1000), and anti-TBK1 (1:1000) were from Cell Signaling Technology. Anti-β-actin (1:1000) (Santa Cruz Biotechnology) was used as housekeeping protein. Goat anti-mouse or goat anti-rabbit horseradish peroxidase (HRP)-conjugated antibodies (1:10,000) (Santa Cruz Biotechnology) were used as the secondary antibody. The membranes were visualized using the ECL system (Bio-Rad) and the expression levels of protein were normalized to actin protein expression levels. Western blots source data provided in the Source Data file.

**Flow cytometry.** Flow cytometry was performed on a BD Canto II flow cytometer and analyzed using the FlowJo software (BD Biosciences). A list of antibodies used here was summarized in Table S2. For intracellular staining of IFN-γ, fresh isolated cells are treated with phorbol 12-myristate 13-acetate/ionomycin cocktail according to the manufacturer's specification (BioLegend). The cells were then washed, stained with antibodies against CD3, CD4, and CD8α, fixed with fixation buffer and subsequently stained intracellularly with antibodies against IFN-γ in Intracellular Staining Permeabilization Wash Buffer (BioLegend). Doublets and debris of dead cells were excluded before various gating strategies were applied. Gates and quadrants were set based on isotype control staining, and the mean fluorescence intensity (MFI) values are calculated by minus the MFI of isotype control antibodies.

**ELISA assay.** Cytokines in cell culture medium, blood samples, or lung tissues were analyzed with ELISA Max Deluxe Sets from BioLegend by following the manufacturer's instructions. After adding the HRP substrates, optical densities were determined at a wavelength of 450 nm in an ELISA plate reader (Bio-Rad).

**Liver enzyme assay.** Serum aspartate transaminase activity (Enzychrom Aspartate Transaminase Assay Kit, BioAssay Systems, Haymard, CA) and alanine transaminase activity (Enzychrom Alanine Transaminase Assay Kit, BioAssay Systems) were performed following the manufacturer's instructions.

**In vitro DC cross-presentation assay.** BMDCs were generated and phenotyped by determining the expression of CD11c. Semi-confluent B16-OVA cells were irradiated with 0 or 20 Gy of a single dose and then immediately seeded into 24-well plate at $1 \times 10^5$ cells per well and cultured for 72 h. Wells were washed twice and $2 \times 10^4$ BMDCs were added and cultured in the presence of NP-cGAMP, free cGAMP, or controls at 37 °C for 18 h. OVA$_{257-264}$ presented with MHC-I on the cell surface was detected by anti-H-2Kb bound to SIINFEKL-PE-Cy7, an antibody that specifically recognizes OVA peptide SIINFEKL bound to H-2Kb of MHC-I by FACS.

**In vitro CD8 T cell priming assay.** CD8$^+$ T cells were isolated from spleens of C57BL/6-Tg (TcraTcrb)1100Mjb/J (OT-I) mice (Jackson Laboratory) by magnetic separation (STEMCELL Technologies) according to the manufacturer's instructions. The purity of CD8$^+$ T cells was >95%. A total of $1 \times 10^5$ cells CD8$^+$ T cells were added into the mixture of BMDCs with the B16-OVA cells pretreated with/without IR in the presence of NP-cGAMP or controls, as described above. After 18 h, cell culture supernatants were collected and measured for IFN-γ content as a surrogate of activation of tumor-specific CD8$^+$ T cells, by ELISA assay.

**In vivo lung metastasis models.** All animal experiments complied with all relevant ethical regulations for animal testing and research and were performed with approved of the Institutional Animal Care and Use Committee at the Wake Forest University School of Medicine. For the B16-OVA lung metastasis model, C57BL/6 mice (6–8 weeks, female:male at 1:1; Charles River Laboratories, Wilmington, MA) were injected intravenously with $2 \times 10^5$ B16-OVA cells. Five days later, the mice developed multifocal metastases on both lungs. For the 4T1-luc lung metastasis model, 4T1-luc cells were injected orthotopically into the right fourth mammary fat pad of female BALB/c mice (8–10 weeks; Charles River). When the tumor volume reached ~500 mm$^3$, the primary tumor was surgically removed in a subset of animals. Development of lung metastases was confirmed before treatment by BLI or MRI. In vivo mechanistic studies and treatment were subsequently conducted on the mice, as described below in details.

**Inhalation of aerosolized PS-coated NPs.** A clear plastic box with a wire-netting floor was used for inhalation treatment. Aerosol was generated via a medical-grade nebulizer attached by medical tubing to the animal chamber (Supplementary Fig. 4). Three unanesthetized animals were placed in the sealed chamber at each time and exposed to aerosol at an air flow rate of 7 L/min for 28 min, during which ~5 mL of solution loaded in the nebulizer were aerosolized. The MMAD and GSD of aerosolized particles and aerosol concentration were measured using a TSI 3321 Aerodynamic Particle Sizer Spectrometer. For in vivo biodistribution study, DiR- or Rhod-b-labeled PS-coated NP-CaP (DiR 660 μg; Rhod-b 80 μg) in 5 mL PBS was loaded into the nebulizer. For treatment, NP-cGAMP (37 μM cGAMP) in 5 mL PBS was loaded into the nebulizer.

**In vivo biodistribution and quantification of NP-cGAMP.** BALB/c healthy mice and mice with 4T1-luc lung metastasis were placed in the chamber to inhale DiR-labeled PS-coated NP for 28 min. The animals were sacrificed at 1, 24, and 48 h ($n = 3$/time) and major organs were dissected and ex vivo imaging was conducted using an IVIS® Lumina system (Caliper Life Sciences). The signal intensity was analyzed using the Living Image® 3.1 software. Immediately after ex vivo imaging, the lung tissues were preserved and sectioned for immunofluorescence microscopy. Cryosections (6 μm thick) were co-stained with anti-mouse CD11c-FITC (1:200; BioLegend, N418) and anti-luciferase (1:500; Sigma-Aldrich, L0159), followed by cy3-anti-rabbit secondary antibody (1:800; Jackson Immuno) and observed using a fluorescence microscope. DiR signals were recorded and merged with the CD11c image and the luciferase-stained image of the same field. DiR$^+$ cells in lung tissues and TDLNs were also assessed by FACS to quantify tissue concentrations of PS-coated NPs, the 4T1-Luc lung metastasis mice were sacrificed at 1, 24, and 48 h after inhaling Rhod-b-labeled PS-coated NP ($n = 3$/time). Major organs and blood were collected for HPLC analyses[56]. For Rhod-b detection, HPLC grade acetonitrile and water (90:10, v/v) with 0.1% trifluoroacetic acid (TFA) was used as mobile phase A and HPLC grade tetrahydrofuran with 0.1% TFA was used as mobile phase B. The mobile phase was delivered at 1.0 mL/min with A:B = 7:3 at 30 °C. The fluorescence detector was set at 540 nm for excitation and 590 nm for emission, and linked to ChemStation LC 3D system for data analysis. Linear calibration curves for concentrations in the range of 0.004–100 ng/μL were plotted using the peak areas by linear regression analysis and concentration of Rhod-b in each sample was determined. Based on HPLC measurements of Rhod-b-NPs in lungs, the concentration of cGAMP deposited in lungs 1 h after inhalation was determined. The fluorescent dye-labeled cGAMP (Biolog Life Science Institute GmbH & Co. KG) was also used to quantify cGAMP in tumor-bearing lungs after inhalation. For the cGAMP detection, HPLC grade acetonitrile with 0.1% TFA was used as mobile phase A and HPLC grade water with 0.1% TFA was used as mobile phase B. The mobile phase was delivered at 1.0 mL/min with A:B = 5:5 at 30 °C. The fluorescence detector was set at 494 nm for excitation and 517 nm for emission, and linked to ChemStation LC 3D system for data analysis.

**In vivo VITAL cell killing assay.** Spleen cells from naive C57BL/6 mice were isolated and pulsed with/without OVA$_{257-264}$ for 2 h in complete medium. The non-pulsed and OVA-pulsed cells were then labeled with high (0.5) or low (0.05) CFSE, respectively, in serum-free medium for 15 min. Equal numbers ($1 \times 10^7$) of CFSE$^{high}$ (non-pulsed) and CFSE$^{low}$ (OVA-pulsed) cells were mixed and injected intravenously into the B16-OVA mice of different treatment groups (on day 18 post tumor implant). After 16 h, blood was collected and subjected to flow cytometry analysis. The number of CFSE$^{high}$ and CFSE$^{low}$ cells were determined and used to calculate the percentage of OVA peptide-specific cell killing based on the following equation:

$$\text{Percentage of specific killing} = (1 - \text{non-transferred control ratio/experimental ratio})100.$$

(1)

**Irradiation.** Animals were anesthetized with inhaled 3% isoflurane and positioned with the lung 50 cm below the aperture of an X-RAD 320 orthovoltage irradiator (Precision X-ray). Fractionated IR (8 Gy × 3 daily fractions) was delivered at a rate of 176 cGy/min (300 kVp voltage and 10 mA current) through a custom-fabricated lipowitz alloy shield. The home-made half circle-shaped lipowitz alloy shield with a radius of 5 mm was placed 3.5 cm above the animal. This shield yields a semi-circular field of ~1.08 cm in diameter, allowing a specified area of the right lung to be irradiated while minimizing the radiation dose to important mediastinum and tissues outside the right lung. To ensure good reproducibility, animal landmarks such as the sternum and ribs were used for initial radiation, the light field of which was marked on the animal, where the radiation light field was aligned with the 10 mm line on the animal for the second and third IR (Supplementary Fig. 8).

**In vivo experimental treatment.** For the B16-OVA lung metastases, after confirming visible multifocal lung lesions on the surface of both lungs by sacrificing randomly selected animals on day 5, the mice ($n = 6$/group) were randomly grouped and treated as follows: (i) PBS (inhalation), (ii) NP-CTR (inhalation of NP-2'5'-GpAp), (iii) IR (8 Gy × 3 to the right lung), (iv) inhalation of NP-cGAMP, (v) NP-cGAMP + IR, (vi) NP-cGAMP + IR + Clod (inhalation of NP-Clod), (vii) NP-cGAMP + IR + anti-CD4, and (viii) NP-cGAMP + IR + anti-CD8α.

Inhalation of NP-cGAMP or controls occurred 24 h after each IR for a total of 3 doses. To deplete pulmonary APCs, the mice inhaled NP-Clod (200 μg) 6 h before each NP-cGAMP for 3 doses. For depleting CD4+ or CD8+ T cells, 400 μg anti-mouse CD4+ antibody (BioXCell, cloneGK1) or anti-mouse CD8α antibody (BioXCell, clone2.43) was injected i.p. one day before treatment and repeated 7 days later. The mice were sacrificed for enumeration of metastatic lung foci on day 18. Both the left and right lungs were evaluated under a dissecting microscope and the blinded quantification conducted. For the 4T1-luc model, a subset of animals was subjected to surgical removal of the primary tumor, while the primary tumor in the other subset of mice was kept without resection, as described above. Lung metastases were visualized by BLI and MRI and confirmed by ex vivo examination before treatment in both studies. The mice ($n = 8$/group) were then randomly grouped and treated as follows: (1) without primary tumor: (i) PBS, (ii) NP-CTR, (iii) IR (8 Gy × 3 to the right lung), (iv) inhalation of NP-cGAMP, (v) NP-cGAMP + IR, (vi) NP-cGAMP + IR + Clod; (2) with primary tumor: (i) PBS, (ii) inhaled NP-cGAMP + IR to the right lung + intratumoral PBS, (iii) inhaled NP-cGAMP + IR to the right lung + intratumoral NP-cGAMP (1 μg × 2; first dose, post IR, but 24 h before the last inhalation; second dose, 24 h post the last inhalation), (iv) inhaled NP-cGAMP + IR to the right lung + intratumoral NP-cGAMP (5 μg × 2). Lung metastases burden was monitored longitudinally by BLI and MRI, while a caliper was used for measuring the primary tumor volume. Survival of the mice was followed for up to 150 days. For tumor re-challenge study, long-term surviving mice from the combination treatment group (56 days) were implanted with 4T1-luc cells ($10^6$) into the contralateral mammary fat pad. Additional control mice were implanted to confirm tumor growth.

**BLI of initiation and development of 4T1-luc lung metastases**. Longitudinal BLI was conducted after orthotopic implantation of 4T1-luc breast cancer. The mice were anesthetized with 2% isoflurane inhalation and injected i.p. with 150 mg/kg D-luciferin. BLI was acquired 10 min later using the IVIS Lumina Imaging System (Caliper Life Sciences). Data were quantified with the Living Imaging software by using absolute photon counts (photons/s/cm$^2$/Sr) in a region of interest, manually drawn to outline the BLI signal of the chest.

**MRI follow-up of lung metastases volume**. MRI was conducted on a 7 T Bruker BioSpec small animal scanner (Bruker Biospin, Rheinstetten, Germany). The imager of MRI was blinded to the group allocation. MRI was initiated on day 18 and followed on days 32 and 39. The mice were anesthetized with isoflurane (3% induction, 1.5% maintenance). Respiration was monitored with a respiratory bulb under the chest and a SHARPII animal monitoring system was used for respiratory gating. Anatomical T2-weighted imaging was conducted using a RARE sequence with TR/TE = 1600/23 ms; ETL: 8; NSA: 8; matrix size: 128 × 128. Individual tumor volumes were measured on T2-W images by manually outlining the enhancing portion of the mass on each image and the total of lung metastases volume for each animal was the sum of individual volumes.

**Histology and immunohistochemistry**. H&E staining was performed on cryo-sections (10 μm) of different tissues, including normal heart, lung, liver, and spleen, as well as lung metastases-bearing lungs. For immunohistochemical staining of tumor-infiltrating lymphocytes, cryosections (10 μm) of 4T1-luc lung metastases-bearing lung tissues obtained from the above treatment group on day 22 (24 h after the last inhalation) were immunostained with anti-mouse CD8α (1:500; BioLegend) or anti-mouse FoxP3 antibody (1:500; BioLegend), followed by HRP-conjugated goat anti-rat secondary antibody (1:500; Jackson Immuno). The sections were then developed with DAB Kits (3,3′-diaminobenzidine; Vector Laboratories) and counterstained by hematoxylin.

**Statistical analysis**. Statistical analysis was performed using Microsoft Excel and Prism 7.0 (GraphPad). Data were presented as mean ± SD. Statistical significance was determined by Student's $t$ test. All $t$ tests were one-tailed and unpaired and were considered statistically significant if $p < 0.05$. The survival assay was analyzed using a log-rank test and considered statistically significant if $p < 0.05$.

**Reporting summary**. Further information on research design is available in the Nature Research Reporting Summary linked to this article.

## Data availability

The authors declare that data supporting the findings of this study are available within this article and its Supplementary Information, and all additional data are available from the corresponding author on reasonable request. A Source Data file, including the source data for Figs. 1–8 in the main text and Supplementary Figs. 1–5, 9, and 11–16, has been provided.

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

## Acknowledgements

We are grateful to Dr. Craig Hamilton (BME, Wake Forest) for his assistance in MR imaging of lung, Dr. J. Daniel Bourland (Radiation Oncology, Wake Forest) for his guidance on animal IR, Dr. David R. Soto Pantoja (Wake Forest) for providing 4T1-luc breast cancer cells, and Dr. Aaron J. Prussin and Dr. Linsey Marr (Virginia Tech) for their assistance in aerosol characterization. We also thank Dr. Ravi Singh (Cancer Biology, Wake Forest), Dr. Frank Marini (Regenerative Medicine, Wake Forest), and Dr. Cristina Furdui (Molecular Medicine, Wake Forest) for technical and collegial support. The research is supported in part by Wake Forest Comprehensive Cancer Center P30 CA01219740 and the Wells Fargo Scholar Program. A.A.H. is also supported by a grant from the Office of Medical Research at the Department of Veterans Affairs.

## Author contributions

D.Z. and Y.L.[1] conceived and designed the study. D.Z. and Y.L.[1] designed experiments. Y.L.[1], W.N.C, L.W., Y.L.[2] and D.Z. performed or assisted with experiments. Y.L.[1], W.N.C, L.W., Y.L.[2], A.A.H., W.P. and D.Z. analyzed the data. D.Z. and Y.L.[1] wrote the manuscript with contributions from W.N.C and W.P. All authors approved the manuscript.

## Competing interests

The authors declare no competing interests.

## Additional information



