## [Peer Review File · Nature Communications]

Editorial Note: This manuscript has been previously reviewed at another journal that is not operating a transparent peer review scheme. This document only contains reviewer comments and rebuttal letters for versions considered at *Nature Communications*

REVIEWERS' COMMENTS:

Reviewer #3 (Remarks to the Author):

The authors have done a very good job in addressing the concerns raised in previous reviews for [REDACTED]. The manuscript is now appropriate for publication in Nat. Commun. However, the authors should tone down the translational potential of the present approach in view of the 8 Gy X 3 X-ray dose used and the poor track record of immunotherapy (other than PD-1/PD-L1 antibodies) in the clinic. Many immune modulators have been published in CNS journals and some of them have been tested in the clinic. The results to date have been disappointing with the exception of PD-(L)1 antibodies and to a lesser extent, CTLA-4 antibody from BMS. STING agonist is no exception. The X-ray dose is too high in animal models and with orthovoltage X-rays. The authors should look up some recent publications in nanoparticle-based radioenhancers that elicit strong antitumor effects with less than 1 Gy/fraction of X-ray doses (for fewer than 5 fractions). The least impressive one, NBTXR3, was recently approved as a medical device in Europe. It is important for the authors to place the present work in the right context, particularly with respect to clinical translation.

Reviewer #4 (Remarks to the Author):

I believe this is a very interesting paper where inhalation of STING activated nanoparticle together with focussed radiotherapy modulates local tumor immunity to control metastasis formation. The authors addressed many questions solidly and I think this paper is of added value to the community to be published

REVIEWERS' COMMENTS:

Reviewer #3 (Remarks to the Author):

The authors have done a very good job in addressing the concerns raised in previous reviews for [REDACTED]. The manuscript is now appropriate for publication in Nat. Commun. However, the authors should tone down the translational potential of the present approach in view of the 8 Gy X 3 X-ray dose used and the poor track record of immunotherapy (other than PD-1/PD-L1 antibodies) in the clinic. Many immune modulators have been published in CNS journals and some of them have been tested in the clinic. The results to date have been disappointing with the exception of PD-(L)1 antibodies and to a lesser extent, CTLA-4 antibody from BMS. STING agonist is no exception. The X-ray dose is too high in animal models and with orthovoltage X-rays. The authors should look up some recent publications in nanoparticle-based radioenhancers that elicit strong antitumor effects with less than 1 Gy/fraction of X-ray doses (for fewer than 5 fractions). The least impressive one, NBTXR3, was recently approved as a medical device in Europe. It is important for the authors to place the present work in the right context, particularly with respect to clinical translation.

We thank the reviewer for positive comments and constructive suggestions. The suggestions, in terms of radiation dose and the potential for a combination with immune checkpoint inhibitors, in particular PD-1/L1 Abs, are well taken, and we are testing the combination of PD-L1 Ab with our inhalation strategy to treat primary lung cancer in one of our ongoing projects. We are also exploring different radiation doses including the low dose regimen in these studies.

Reviewer #4 (Remarks to the Author):

I believe this is a very interesting paper where inhalation of STING activated nanoparticle together with focused radiotherapy modulates local tumor immunity to control metastasis formation. The authors addressed many questions solidly and I think this paper is of added value to the community to be published.

We appreciate the reviewer's comments.